# Fine-Tuning Heat Stress Algorithms to Optimise Global Predictions of Mass Coral Bleaching

**Liam Lachs [1,\*], John C Bythell [1], Holly K East [2], Alasdair J Edwards [1], Peter J Mumby [3,4], William J Skirving [5,6], Blake L Spady [5,6] and James R. Guest [1]**

[1] School of Natural & Environmental Sciences, Newcastle University, Newcastle upon Tyne NE1 7RU, UK; john.bythell@newcastle.ac.uk (J.C.B.); alasdair.edwards@newcastle.ac.uk (A.J.E.); james.guest1@newcastle.ac.uk (J.R.G.)

[2] Department of Geography and Environmental Sciences, Northumbria University, Newcastle upon Tyne NE1 7RU, UK; holly.east@northumbria.ac.uk

[3] Marine Spatial Ecology Lab, School of Biological Sciences, University of Queensland, St Lucia, QLD 4072, Australia; p.j.mumby@uq.edu.au

[4] Palau International Coral Reef Center, Koror 96940, Palau

[5] Coral Reef Watch, National Oceanic and Atmospheric Administration, College Park, MD 20740, USA; william.skirving@noaa.gov (W.J.S.); blake.spady@noaa.gov (B.L.S.)

[6] ReefSense Pty, Ltd., P.O. Box 343, Aitkenvale BC, Aitkenvale, QLD 4814, Australia

\* Correspondence: l.lachs2@newcastle.ac.uk

**Abstract:** Increasingly intense marine heatwaves threaten the persistence of many marine ecosystems. Heat stress-mediated episodes of mass coral bleaching have led to catastrophic coral mortality globally. Remotely monitoring and forecasting such biotic responses to heat stress is key for effective marine ecosystem management. The Degree Heating Week (DHW) metric, designed to monitor coral bleaching risk, reflects the duration and intensity of heat stress events and is computed by accumulating SST anomalies (HotSpot) relative to a stress threshold over a 12-week moving window. Despite significant improvements in the underlying SST datasets, corresponding revisions of the HotSpot threshold and accumulation window are still lacking. Here, we fine-tune the operational DHW algorithm to optimise coral bleaching predictions using the 5 km satellite-based SSTs (CoralTemp v3.1) and a global coral bleaching dataset (37,871 observations, National Oceanic and Atmospheric Administration). After developing 234 test DHW algorithms with different combinations of the HotSpot threshold and accumulation window, we compared their bleaching prediction ability using spatiotemporal Bayesian hierarchical models and sensitivity–specificity analyses. Peak DHW performance was reached using HotSpot thresholds less than or equal to the maximum of monthly means SST climatology (MMM) and accumulation windows of 4–8 weeks. This new configuration correctly predicted up to an additional 310 bleaching observations globally compared to the operational DHW algorithm, an improved hit rate of 7.9%. Given the detrimental impacts of marine heatwaves across ecosystems, heat stress algorithms could also be fine-tuned for other biological systems, improving scientific accuracy, and enabling ecosystem governance.

**Keywords:** marine heatwaves; sea surface temperature; mass coral bleaching; algorithm optimisation; spatiotemporal Bayesian modelling; R-INLA

## 1. Introduction

Anthropocene marine heatwaves are becoming increasingly intense, more frequent and longer lasting due to climate change [1,2]. These anomalous heat stress events can have severe implications for a range of marine biota, e.g., influencing shifts in zooplankton communities, declines in key groups such as krill [3–5], die-offs and reproductive failures of sea-birds [6–8], marine mammal strandings [8] and mass coral bleaching and mor-

tality events [9]. While surveying in situ ecosystem responses to climate change disturbances is essential to assess impact, it is also very costly. Accurate monitoring of ecosystem stress remotely and at scale is therefore crucial for effectively managing marine ecosystems and accurately predicting the impacts of climate change on marine biota. While satellite-based remote monitoring and forecasting programmes have been implemented across various biological contexts, we focus this study specifically on remote monitoring and forecasting of coral bleaching. Coral reefs are highly productive ecosystems that provide habitat to over a million marine species and essential ecosystem services (e.g., coastal protection, food, fisheries and tourism livelihoods) to hundreds of millions of people, estimated to be worth over 350,000 USD ha$^{-1}$ yr$^{-1}$ globally [10,11]. These ecosystems are increasingly faced with mass coral bleaching and mortality events [12]. The process of coral bleaching involves a breakdown in the symbiosis between coral hosts and their endosymbiotic phototrophic algae and can ultimately lead to full or partial colony mortality [13] and sub-lethal effects such as reduced growth [14]. Coral bleaching is a stress response with a variety of triggers (e.g., anomalous temperature, both high and low; anomalous increases in solar insolation; anomalous salinity, both high and low; reduction in water quality; and disease; [15]). Episodes of mass coral bleaching occur across large spatial scales, affect numerous coral taxa and can destroy entire healthy reefs within months. Pantropical mass bleaching events are becoming recurrent and are caused by the widespread increasing incidence of marine heatwaves under climate change [12,16,17].

Over the past two decades, the National Oceanic and Atmospheric Administration's (NOAA) Coral Reef Watch (CRW) programme has developed a suite of tools for monitoring coral bleaching risk using satellite-based sea surface temperature (SST) products. Specifically, the Degree Heating Week (DHW) metric is used as an indicator of heat stress levels sufficient to induce coral bleaching. DHW is computed as the accumulation of positive temperature anomalies (HotSpot) above a hypothesised coral bleaching stress temperature (i.e., 1 °C above the maximum of monthly means SST climatology—MMM) over the previous 12 weeks [18,19]. The DHW algorithm was designed in the 1990s, and the HotSpot threshold (1 °C above MMM) and accumulation window (12 weeks) were chosen based on field and experimental evidence from Panama and the Caribbean [20,21]. Reflecting the technological advancements in remote sensing capabilities since then, the SST and DHW products have increased in spatial resolution (50 to 5 km) and temporal resolution (twice weekly to daily) [22]. Despite these improvements, there has not yet been a corresponding revision of the HotSpot threshold and accumulation window used in the operational DHW algorithm.

Alternate DHW algorithms have been applied to evaluate associations between heat stress and coral bleaching, mostly at local or regional scales [18,23–29]. Particularly for weak marine heatwaves associated with coral bleaching, computing DHWs with a lower HotSpot threshold has proven useful for monitoring bleaching impacts and severity [23–25]. Evidence also suggests that using a shorter accumulation window in the DHW algorithm can improve coral bleaching predictions in some cases [26,30]. An optimisation study in which numerous DHW algorithms are tested against a global coral bleaching dataset could provide the scientific basis necessary to revise the operational DHW metric. Recently, [30] showed that altering the HotSpot threshold and accumulation window can improve global coral bleaching prediction skill, based on weather forecasting techniques that predict bleaching events (yes or no) depending on whether DHWs exceed a certain threshold or not. [30] used DHWs computed from the Optimum Interpolation SST (OI-SSTv2) and coral bleaching records from a summative dataset of 100 well-studied coral reefs [9]. However, there is a mismatch in spatial scale between these two datasets; the SST data were extracted from 0.25-degree grid cells (~770 km$^2$ at the equator), while the area extent of each reef in the bleaching dataset ranged from 2 (Southwest Rocks, Australia, and St. Lucia, South Africa) to over 9000 km$^2$ (Northern Great Barrier Reef, Australia). Accordingly, there are potential mismatches between DHW values and bleaching data in their study. As such, there is a pressing need to apply a more comprehensive DHW

optimisation study to a global dataset of direct bleaching observations and DHWs derived from a higher resolution SST dataset.

To construct a global coral bleaching model based on environmental covariates, predictions should account for spatial and temporal dependencies. For example, corals in certain geographic regions are likely to respond to heat stress with higher levels of coral bleaching (e.g., areas influenced by the El Niño Southern Oscillation) [31,32] and stress responses are likely to change through time due to coral adaptation and assemblage turnover [33,34]. From a statistical standpoint, spatiotemporal uncertainties in the bleaching–environment relationship must be accounted for to ensure that bleaching predictions are not just artefacts of spatial or temporal patterns in unmeasured variables. A number of studies modelling coral bleaching globally as a function of environmental covariates have assumed that the uncertainty of this relationship is spatiotemporally constant [30,35]. This assumption is unlikely to be true for coral bleaching responses, given the potential for coral adaptation [36,37] and the extent to which post-disturbance turnover can alter the composition of the coral assemblage [34] and therefore its tendency to experience subsequent coral bleaching. To address the spatial (but not temporal) issues, [38] introduced a Bayesian mixed modelling approach that explicitly resolved spatial variability in the uncertainty of bleaching–environment relationships. This was achieved by treating ecoregion and site as hierarchical random effects, but this comes at the cost of a slow runtime, an issue further compounded by implementing these models via Monte Carlo Markov chains (MCMC) which run iteratively and slowly [39]. Given these issues, such an approach would not be appropriate for a coral bleaching optimisation study that aims to test hundreds of DHW algorithms (i.e., hundreds of statistical models) whilst also accounting for spatial and temporal dependencies, since such a study would require a prohibitively large amount of computing resources.

This study seeks to offer a potential revision to the operational NOAA DHW metric with a view to improving its ability to predict mass coral bleaching. This will require a suitable methodology that is robust to spatiotemporal correlated uncertainties and runs with a reasonable computational speed. Here, we apply an alternative approach to modelling bleaching–environment relationships based on integrated nested Laplace approximation (INLA), which explicitly solves spatial and temporal uncertainties with much greater computational speed than MCMC [39]. We aim to optimise two DHW algorithm parameters, the HotSpot threshold (testing levels from MMM −4 to +4 °C) and the accumulation window (testing levels from 2 to 52 weeks) to improve coral bleaching predictions globally whilst still addressing the common issue of spatial and temporal dependencies. We achieved this by combining recently developed Bayesian hierarchical modelling techniques using INLA with a streamlined parallel computing workflow on a high-performance computing cluster called "The Rocket". This allowed hundreds of spatiotemporal INLA models to be run in a short time frame (i.e., hours instead of weeks, as would be the case using MCMC).

## 2. Materials and Methods

### 2.1. Coral Bleaching Data

The optimisation study presented here was based on a global dataset of 37,871 bleaching survey records from published and unpublished scientific sources spanning from 1969 to 2017 [40–42]. Bleaching estimates were quantified by a wide range of surveying methods, including aerial surveys, line-intercept transects, belt transects, quadrats, radius plots, rapid visual assessments (e.g., manta tows), ad hoc estimates and interviews with stakeholders. Since data were collected by hundreds of observers globally over several decades, data collection protocols for these different general methods are not standardised.

The original dataset underwent four layers of filtering a priori to ensure its suitability for analyses. (1) Data were first filtered for errors. This excluded observations that did not

have a recorded month or year, as well as observations in which the coordinates provided did not correspond with a coral reef location (5562 observations excluded). (2) Data were removed if the survey date fell outside the period of peak thermal exposure for that year. As, for the purpose of this study, we are only interested in coral bleaching that results from thermal stress (i.e., not bleaching due to cold stress, nutrient enrichment, etc.), instances of bleaching that cannot be linked to the period of peak thermal exposure may not accurately reflect the status of heat-induced bleaching for that year and location. We defined the period of peak thermal exposure as the month prior to the month of MMM up to three months after the month of MMM. For example, if the month of MMM was February for a certain location, only observations from January to May were included. Further, we ensured that the observation was not made before the date of maximum DHW in that year (19,292 observations excluded). (3) To account for different sampling protocols in records of percentage bleaching, we computed bleaching as a binary variable. Bleaching estimates were reported as means, ranges or broad categories. First, we summarised these as representative minimum and maximum percentages. Then, the absence of ecologically significant bleaching was defined as having a maximum estimation of 10% bleaching or less, while the presence of ecologically significant bleaching was defined as having a minimum estimation of 20% bleaching or greater. Observations in which the maximum estimation exceeded 10% while the minimum estimation remained below 20% were filtered out to reduce the chance of misrepresenting bleaching and non-bleaching observations (Figure S1) (1452 observations excluded). (4) Lastly, to account for spatiotemporal patchiness a priori, we only retained years which had greater than 100 independent observations, had a qualitatively even global distribution and were not temporally isolated (i.e., all proceeding years also needed to meet the previous two criteria). This resulted in removal of all data before 2003. Despite having 345 bleaching records in 2002, all data from this year were removed as over 80% of records were from the Great Barrier Reef region alone (1185 observations excluded). The resulting dataset included 10,380 unique observations between 2003 and 2017, with >171 observations per year and sufficient spatial representation for each year (Figure 1A).

Accumulated heat stress (see Figure 1B for a DHW example) is considered to be the mechanism causing mass coral bleaching [43,44], and marine heatwaves typically occur across hundreds to thousands of kilometers on spatial scales of weather systems. The vast majority of bleaching observations in the dataset are associated with mass bleaching events, but despite our filtration process, some bleaching observations will inevitably result from small-scale local heat stress and other non-heat-related factors. Since the models presented in this study are based solely on large-scale accumulated heat stress, the model predictions we present reflect the mechanism of mass coral bleaching (i.e., across large spatial scales and numerous taxa due to heat stress).

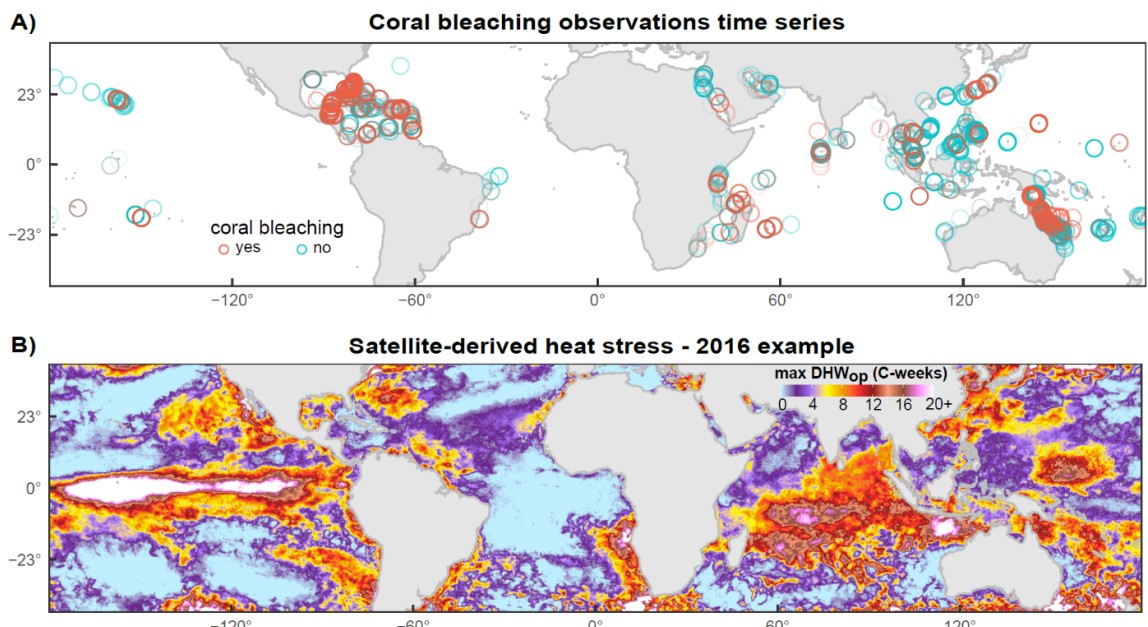

**Figure 1. (A)** Distribution of coral bleaching survey records based on estimates of percentage coral bleaching (<10% = no, >20% = yes), measured at 5724 sites from 84 countries between 2003 and 2017 (N = 10,380) after four layers of a priori filtering (i.e., removal of errors, matching surveys with the period of peak thermal exposure in the year, accounting for inconsistent sampling protocols and accounting for spatiotemporal patchiness). **(B)** World map of maximum levels of heat stress reached in 2016 as shown by the maximum operational Degree Heating Week (DHW_op) metric derived from 5 km resolution satellite-based sea surface temperature data. Marine heatwaves in 2016 caused widespread mass coral bleaching globally. Coral bleaching observations from specific locations and years (**A**) were matched with maximum DHW values from the 5 km grid cell surrounding the observation coordinates (e.g., **B**).

### 2.2. Temperature Data

Heat stress metrics were derived from a combination of CoralTemp v3.1 [19], a gap-free global 5 km daily satellite-based SST dataset from 1985 until present, and the corresponding 5 km MMM climatology dataset [19,45]. MMM reflects the typical upper limit of the summertime monthly mean SST for the centre of the baseline period of CRW Heritage products (1985–1990 + 1993) [45] and is constant through time for specific locations. At each spatially referenced survey record, environmental data were extracted from the 5 km grid cell encompassing that coordinate. These data consisted of a single MMM value and a time series of the daily SST from the start of the pre-survey year until the end of the survey year.

For the operational DHW metric used by NOAA (DHW_op) (Figure 1B), daily HotSpots were calculated as daily positive SST anomalies relative to MMM (1). Time series of daily DHW_op were then computed using the standard NOAA CRW method (2). HotSpots greater than 1 °C were accumulated across a 12-week moving window (84 days inclusive), where $i$ is the date and $n$ is the earliest date of the accumulation window. Each daily HotSpot used in the summation was divided by seven a priori, such that

$$HotSpot_i = SST_i - \text{MMM}, \qquad HotSpot_i \geq 0 \tag{1}$$

$$Operational\ DHW_i = \sum_{n=i-83}^{i} \left( \frac{HotSpot_n}{7} \right), \qquad \text{for } HotSpot_n \geq 1 \tag{2}$$

As an example, consider a 12-week window ending on 1 April for a specific survey location. This window includes only three daily SSTs that exceed the MMM, equivalent to HotSpots of 0.5, 1.4 and 2.8 °C. The DHW_op value for 1 April is the summation of 1.4

and 2.8° each divided by seven, which is 0.6 °C-weeks. The 0.5 °C HotSpot value was not included in the summation as it was below 1 °C [19].

We computed a total of 234 test DHW metrics (DHW_test), each with unique combinations of HotSpot thresholds (9 levels, from −4 to +4 °C relative to MMM) and accumulation windows (26 levels, from 2 to 52 weeks). Unlike the operational metric, HotSpots for DHW_test were calculated relative to the MMM after an adjustment for the specific threshold in question (3). In the operational metric, only HotSpots >1 °C are accumulated; however, in the test metrics, all positive HotSpots are accumulated. Therefore, values of DHW_test are numerically different than DHW_op but are conceptually the same. Time series of daily DHW_test were computed as the accumulation of HotSpots (4), where *i* is the date, *n* is the earliest date of the accumulation window and *j* is the length of the accumulation window in days minus one, such that

$$HotSpot_i = SST_i - \text{MMM} + \text{HotSpot Threshold}, \qquad HotSpot_i \geq 0 \qquad (3)$$

$$Test\ DHW_i = \sum_{n=i-j}^{i} \left(\frac{HotSpot_n}{7}\right), \qquad for\ HotSpot_n \geq 0 \qquad (4)$$

### 2.3. Statistical Approach

The time unit used in the following models is the calendar year. As coral bleaching is more likely at higher levels of heat stress [43], the maximum of daily DHW values was computed from the year of each survey record (see max DHW_op in Figure 1B). Thus, all further reference to DHW metrics relate to the annual maximum summary statistic. Given that the Southern Hemisphere summer starts before the end of the calendar year, there was a chance of misclassifying maximum DHW values. For instance, a maximum DHW on the first or last day of a calendar year will be part of the same heatwave event; however, they will each be assigned to different calendar years. Previously, this has been addressed by adopting different calendars for each hemisphere [44]; however, this was not necessary in the current study since no such instances were present in the dataset. The relative performance of DHW metrics for predicting mass coral bleaching was assessed systematically using the following conceptual framework.

1. For each DHW metric, the association with coral bleaching was tested using a spatiotemporal generalised linear model (GLM) with a Bernoulli error structure using INLA.
2. Sensitivity–specificity analysis was performed on this GLM to optimise predictions, tally model successes and failures and provide metrics for model comparisons.
3. The first two steps were repeated for all DHW_test metrics and DHW_op, resulting in 235 separate GLMs and sensitivity–specificity analyses, each run in parallel on separate Intel Xeon E5-2699 processors via the high-performance computing cluster "The Rocket".
4. Model comparisons were used to determine the best-performing models and hence the optimal HotSpot threshold and accumulation window for predicting coral bleaching globally using DHWs.

### 2.4. Model Formulation

We adopted a spatiotemporal Bayesian modelling approach to predict mass coral bleaching based on DHWs using the R-INLA package (version 21.01.26) [39]. Compared to more commonly used frequentist approaches, Bayesian inference allows uncertainty to be more easily interpreted. Moreover, using R-INLA over other Bayesian tools (e.g., Monte Carlo Markov chains) provides the opportunity to resolve spatiotemporal correlations explicitly and more rapidly [39].

Observations of mass coral bleaching are often spatiotemporally correlated due to large-scale climatic drivers. While basic linear regressions applied to such data ignore

these dependencies and lead to pseudoreplication [46], R-INLA circumvents these issues. In each time point, spatial dependencies are dealt with by implementing the Matérn correlation across a Gaussian Markov random field (GMRF), essentially a map of spatially correlated uncertainty. This is achieved using stochastic partial differential equations (SPDE) solved on a Delaunay triangulation mesh of the study area. The parameters ($\Omega$) that determine the Matérn correlation are the range ($r$—range at which spatial correlation diminishes) and error ($\sigma$). Weakly informative prior estimates of these parameters ($r_0$ and $\sigma_0$) are recommended when implementing the Matérn correlation [47]. Temporal dependencies among these GMRFs are dealt with by imposing a first-order autoregressive process (AR1), defined by the AR1 parameter ($\rho$) (9). This allows for correlations in model residuals through time avoiding pseudoreplication.

To test the effect of DHW metrics on coral bleaching, a triangular mesh (Figure 2) was defined with a maximum triangle edge length of 600 km and a low-resolution convex hull (convex = −0.03) around the study sites to avoid boundary effects (1790 nodes). This mesh was repeated for each year in the time series (26,400 nodes). The probability of coral bleaching for a given observation ($CB_{t,i}$) in a given year ($t$) and location ($i$) was assumed to follow a Bernoulli distribution ($\pi_{t,i}$) using the logit-link function for binary data. Bleaching was modelled as a function of the DHW metric in question (fixed effect: $DHW_{t,i}$) whilst accounting for additional underlying spatiotemporal correlations among bleaching observations (random effect: $v_{t,i}$),

$$CB_{t,i} \sim \text{Bernoulli}(\pi_{t,i}), \tag{5}$$

$$\text{Expected}(CB_{t,i}) = \pi_{t,i}, \tag{6}$$

$$\text{Variance}(CB_{t,i}) = \pi_{t,i} \times (1 - \pi_{t,i}), \tag{7}$$

$$\text{logit}(\pi_{t,i}) = \beta_0 + \beta_1 \times DHW_{t,i} + v_{t,i} + \varepsilon_{t,i}, \tag{8}$$

$$v_{t,i} = \rho \times v_{t-1,i} + u_{t,i}, \tag{9}$$

$$u_{t,i} \sim \text{GMRF}(0, \Omega), \tag{10}$$

$$\varepsilon_{t,i} \sim \text{Normal}(0, \sigma^2), \tag{11}$$

where $\beta_0$ is the intercept, $\beta_1$ is the DHW parameter estimate, $\rho$ is the AR1 parameter, $u_{t,i}$ represents the smoothed spatial effect from the GMRF mesh, elements of $\Omega$ ($r$ and $\sigma$) are estimated from the Matérn correlation and $\varepsilon_{t,i}$ contains the independently distributed residuals. Following the recommendations from [47], we specified weakly informative priors for $r_0$ (2000 km) and $\sigma_0$ (1.15) based on the residual variogram and error from an intercept-only null Bernoulli GLM (Figure S2). We also tested different priors; however, they had a negligible effect on the estimates of any model parameters. To avoid imposing artificial temporal dependencies, we used a non-informative default prior for $\rho$.

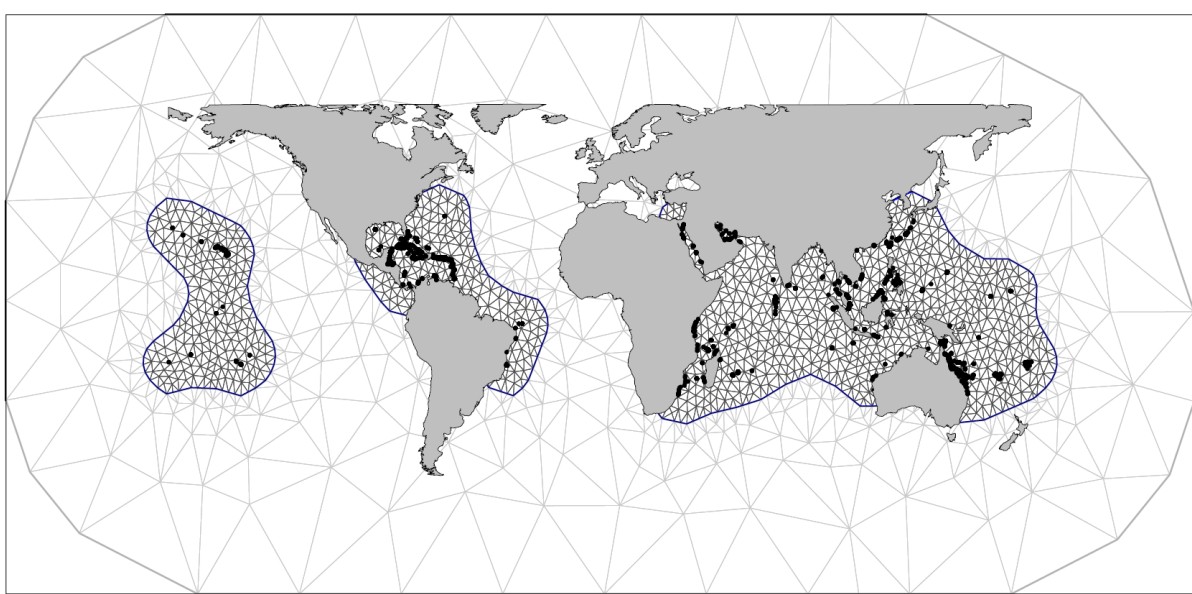

**Figure 2.** Constrained refined Delaunay triangulation mesh of 1790 nodes used for spatial correlation in one timestep. The spatiotemporal correlation over 15 years is computed over 15 such meshes totalling 26,400 nodes. Continents and bleaching survey coordinates (black points) overlay the higher resolution study area (black mesh) and lower resolution convex hull (grey mesh).

### 2.5. Model Validation

Standard model validation steps were conducted for the best-performing GLM and included plotting bleaching observations against fitted values, assessing model residuals for spatiotemporal correlations using maps and variograms and producing a time series of maps showing spatiotemporally correlated uncertainty [48]. The dataset presented here was considerably patchy in both space and time despite prior filtering (e.g., no South Pacific observations in 2003, 2012 or 2013). Patchy data are a pertinent issue in statistics [49] and can have a considerable effect on the estimated model parameters [50], and model selection criteria (e.g., deviance information criterion—DIC) [51]. Thus, to address patchiness beyond basic filtering, we performed a simulation test (Figure S3 and Figure S4). In summary, patchiness did not have an important effect on estimated model parameters (Figure S5), validating the broader model comparison methods and results of the main study. Full details are described in the Supplementary Materials.

### 2.6. Sensitivity–Specificity Analysis

To optimise binary predictions from each Bernoulli GLM, sensitivity–specificity analyses were performed using receiver operating characteristic (ROC) curves in R [52] without considering spatiotemporal dependencies. This method is commonly applied in bioinformatics and medical decision making to determine the performance of binary classifications. Here, sensitivity is defined as the proportion of correctly classified bleaching observations (true positives), and specificity as the proportion of correctly classified non-bleaching observations (true negatives). As a probability cut-off is moved over all possible values, the ROC plot shows the corresponding sensitivity and specificity at each level. The area under the curve (AUC) from each ROC plot reflects the performance of that GLM relative to the perfect predictor (AUC = 1) and can be used for multi-model comparisons based on 95% confidence intervals computed using stratified bootstrap resampling [52]. The hit rate, defined as the proportion of observed bleaching events that were correctly predicted, was also computed at the optimal cut-off level for each model.

*2.7. Model Comparisons*

Model comparisons were based on the Bayesian DIC and two key metrics from the sensitivity–specificity analysis: AUC and hit rate. DIC is a measure of overall model parsimony [48] but is based on both the DHW fixed effect and the spatiotemporal random effect. Therefore, the AUC and bootstrapped confidence intervals were used as the preferred model comparison metric, as they evaluate the overall performance of a binary classifier relative to a perfectly predicting model [52], based on the fixed effect only. Hit rate is an additional metric that allows an easy interpretation of model success.

## 3. Results

*3.1. Model Comparisons*

For predicting coral bleaching based on $DHW_{test}$, we identified (1) a group of worst-performing models, (2) a group of better performing models and (3) a suite of best-performing models. (1) Poor GLM performance was associated with $DHW_{test}$ metrics computed on HotSpot thresholds ≥ MMM + 2 °C or accumulation windows ≥ 22 weeks. This was evident by low AUC values < 0.7 and high DIC values > 7000 (Figure 3, right and upper regions). (2) The remaining GLMs (HotSpot threshold ≤ MMM + 1 °C, accumulation window ≤ 20 weeks) were associated with better coral bleaching predictions (AUC) and model parsimony (DIC) (Figure 3, lower and lower left regions). (3) Finer determination of the best models of this subset was made possible by incorporating sensitivity–specificity uncertainty into model comparisons (Figure 4, 95% bootstrapped confidence intervals). A performance–optima relationship was apparent between the AUC and the HotSpot threshold and accumulation window, whereby peak GLM performance was reached when DHW accumulation windows were 4–8 weeks (Figure 4). When DHW accumulation windows were outside this range (2 weeks or ≥ 10 weeks), the corresponding AUC was significantly lower than the AUC of the best-performing GLMs (Figure 4, blue shaded region). Notably, of all the GLMs that used the same accumulation window (grey and white band groupings, Figure 4), those models applying lower HotSpot thresholds performed better in terms of the AUC and DIC. The 8-week accumulation window resulted in the best overall fit of the AUC and DIC combined (max DIC = 6812). In summary, the suite of best-performing models (in terms of bleaching prediction) applied $DHW_{test}$ metrics based on HotSpot thresholds ≤ MMM and accumulation windows of 4–8 weeks.

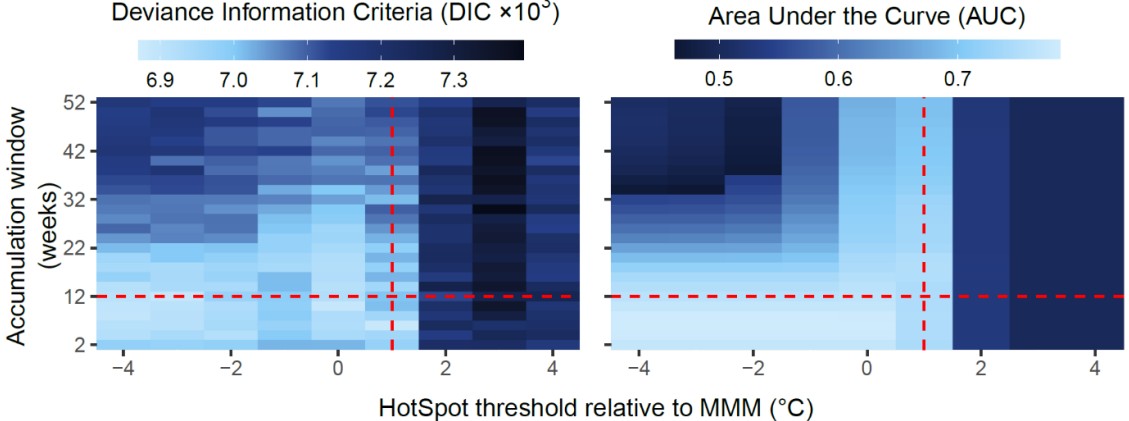

**Figure 3.** Model comparison heatmaps showing the deviance information criterion (DIC) and area under the curve (AUC) for 234 generalised linear models (GLMs) that each predict coral bleaching based on a different $DHW_{test}$ metric. Raster cells represent individual GLMs plotted by HotSpot threshold and accumulation window. The threshold and window used for $DHW_{op}$ are shown by red dashed lines (MMM + 1 °C, and 12-weeks). Results for the $DHW_{op}$ GLM are not shown on the heatmaps (DIC = 6967, AUC = 0.758).

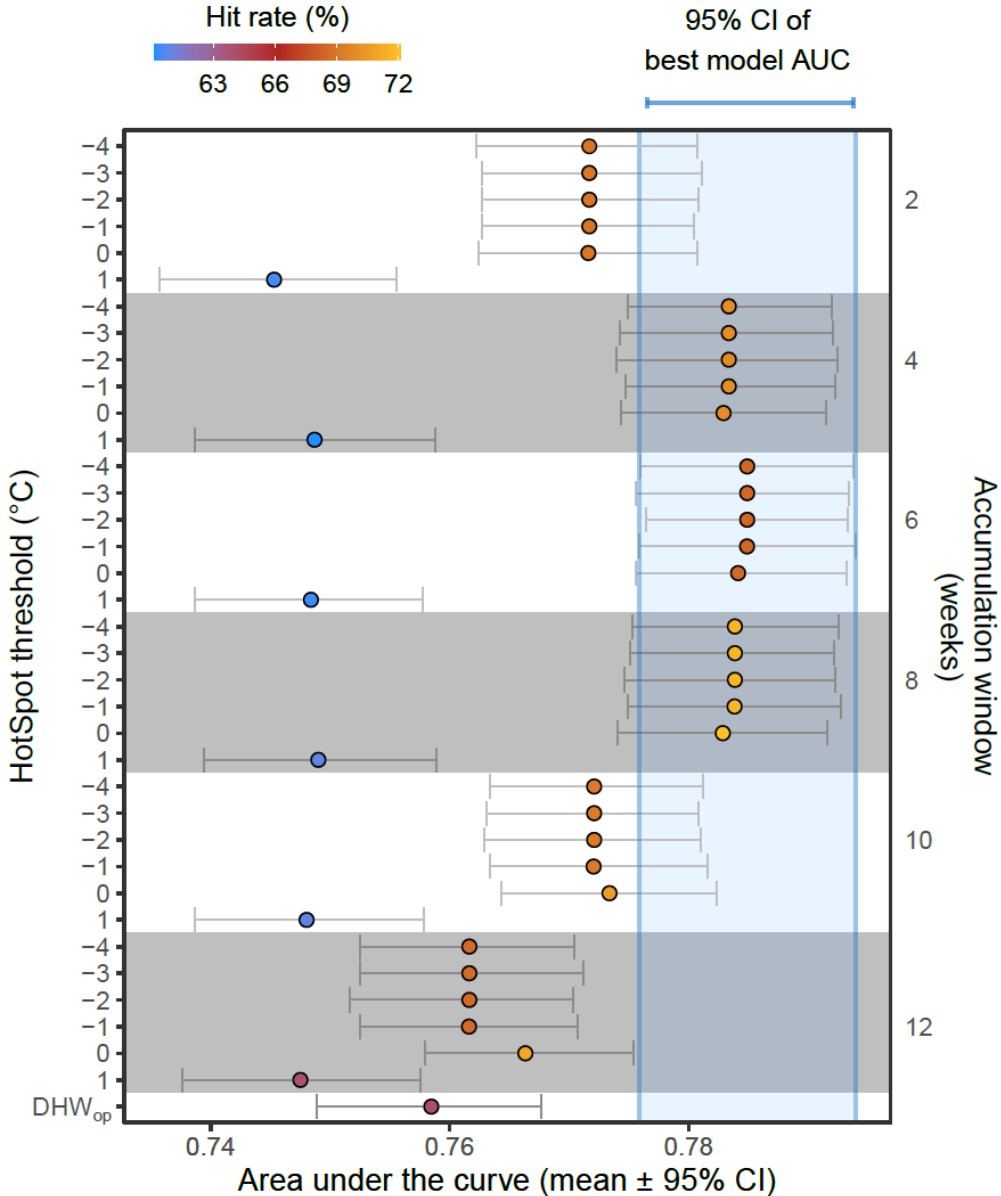

**Figure 4.** Model comparisons accounting for uncertainty in the area under the curve (AUC, derived from sensitivity–specificity analysis) showing the mean and 95% bootstrapped confidence intervals (CI). Each point represents a generalised linear model (GLM) that predicts coral bleaching based on a different DHW$_{test}$ metric, ordered by HotSpot threshold and accumulation window (both increasing downwards). The hit rate (proportion of observed bleaching events correctly predicted) is shown for each GLM (point colour), and the AUC of the best GLM is shown as a blue shaded region. Note the DHW$_{op}$ algorithm is slightly different than the DHW$_{test}$ algorithm (Equations (1)–(4)).

### 3.2. Best Model—Validation

The GLM based on the DHW$_{test}$ metric with a HotSpot threshold of MMM + 0 °C and an accumulation window of 8 weeks (DHW$_{test\text{-}0C\text{-}8wk}$) was a representative of the suite of best-performing models. The probability of bleaching output from this model (based on DHW$_{test\text{-}0C\text{-}8wk}$ and unmeasured spatiotemporally correlated factors) closely matched the observational bleaching record (Figure 5A). Both the fixed effect (DHW$_{test\text{-}0C\text{-}8wk}$) and the random effect (spatiotemporal uncertainty) provided important contributions to predictions of coral bleaching (Figure S7). The sensitivity–specificity analysis reflected the high performance for this model, with an AUC value of 0.783 (Figure 5B). The range parameter

(*r*) of GMRFs showed that drivers of bleaching other than DHW$_{test-0C-8wk}$ were spatially correlated up to 697 km (Figure S6), consistent with the spatial scale of climatic and weather systems. The AR1 parameter (*ρ*) of 0.62 indicated a moderate temporal correlation of uncertainty in predicted coral bleaching (i.e., drivers other than DHW$_{test-0C-8wk}$), meaning that the uncertainty in bleaching predictions in one year is affected by that of the previous year by a factor of 0.62 (Figure S6). This can be seen visually on maps of temporally correlated GMRFs (Figure S7).

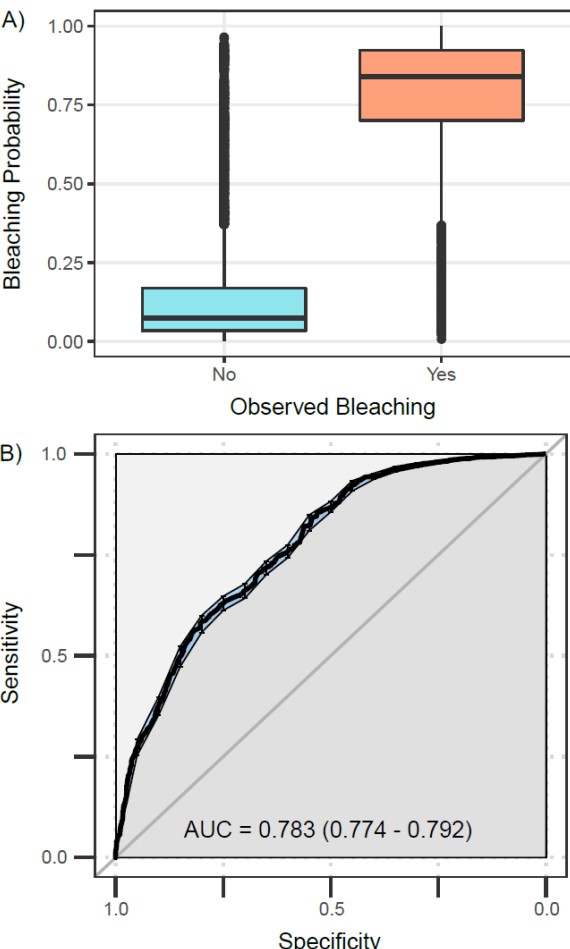

**Figure 5.** Exploration of best-performing GLM which predicts coral bleaching based on DHW$_{test-0C-8wk}$ (HotSpot threshold = MMM, accumulation window = 8 weeks) and spatiotemporal uncertainty. (**A**) Fitted values or bleaching probabilities are shown relative to bleaching observations from the global dataset, showing a clear separation between bleaching and non-bleaching categories. (**B**) Sensitivity–specificity analysis is shown for the same GLM without spatiotemporal uncertainty. Sensitivity is defined as the proportion of correctly classified bleaching observations (true positives), and specificity as the proportion of correctly classified non-bleaching observations (true negatives). Area under the curve (AUC) and bootstrapped 95% confidence intervals (shown in brackets) reflect the distance to a perfectly predicting model (AUC = 1).

### 3.3. Best Model—Understanding Heat Stress

Even though lowering the HotSpot threshold and reducing the accumulation window improved predictions of mass coral bleaching (Figures 3 and 4), the $DHW_{op}$ metric still categorised bleaching observations well. $DHW_{op}$ values were greater for bleaching records than for non-bleaching records (Figure 6). Of the 517 highest heat stress records (>95th percentile: >9.0 °C-weeks), 78% were associated with coral bleaching observations, highlighting the importance of heat stress as a proximate cause of coral bleaching. Such levels of heat stress relate to NOAA CRW Bleaching Alert Level 2. However, in comparison to $DHW_{op}$, the test metric $DHW_{test-0C-8wk}$ showed a higher distribution of heat stress values overall, but lower extreme values (Figure 6). This is due to a lower HotSpot threshold and shorter accumulation window, respectively. This was characterised by fewer DHW values of zero (1 vs. 27%), a higher mean (5.2 vs. 2.5 °C-weeks) and a higher 95th percentile (9.9 vs. 9.0 °C-weeks), but a lower 99th percentile (11.3 vs. 12.5 °C-weeks). The number of bleaching observations associated with a heat stress of 0 was 6 for $DHW_{test-0C-8wk}$ and 122 for $DHW_{op}$. Given that $DHW_{test-0C-8wk}$ had a lower HotSpot threshold, fewer bleaching observations are associated with heat stress values of zero. In other words, reducing the HotSpot threshold increased our ability to predict coral bleaching associated with weak marine heatwaves.

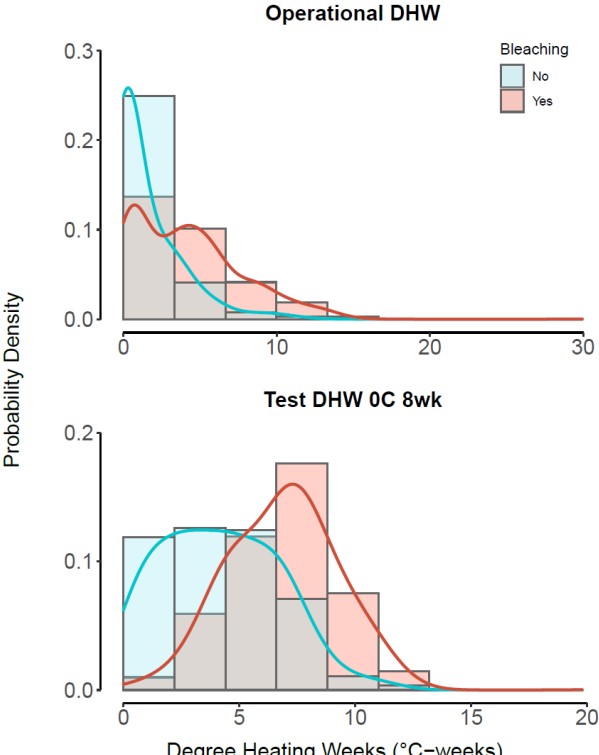

**Figure 6.** DHW distributions for bleaching records (red) and non-bleaching records (blue), shown as histograms and probability density curves. For comparison of different DHW metrics, the operational metric used by NOAA ($DHW_{op}$) is shown alongside one of the best-performing metrics ($DHW_{test-0C-8wk}$), calculated using a lower HotSpot threshold (MMM + 0 °C) and a smaller accumulation window (8 weeks).

## 4. Discussion

Heat stress can have considerable impacts on marine organisms and entire marine ecosystems [53,54]. The DHW metric is a measure of accumulated heat stress widely used to predict mass coral bleaching caused by anomalous temperatures above typical sum-

mertime conditions [35,38,43,44]. The remote-sensed SST products underpinning the operational NOAA DHW metric have improved stepwise over the last two decades [18,19,28,55]; however, there has not yet been a corresponding revision of the HotSpot threshold and accumulation window used in this algorithm. Here, we developed 234 different DHW algorithm variants each with a different HotSpot threshold and accumulation window. We assessed the performance of these $DHW_{test}$ metrics for predicting mass coral bleaching globally. Compared to $DHW_{op}$, it was possible to improve the coral bleaching hit rate by up to 7.9% by using different HotSpot thresholds and accumulation windows, equating to an additional 310 correctly predicted bleached reefs out of a total of 3895 (also linked to an increased false negative rate of 3%). While temperature may be the dominant environmental variable associated with mass coral bleaching events, other variables also have a role. The authors of [15] showed that variations in light leading up to a bleaching event can affect DHW accuracy. By shortening the accumulation window for DHW, we potentially increased the effect of light relative to temperature, which could easily account for an additional 3% of false negatives. As a number of observations of bleaching in the dataset were likely caused by mechanisms other than accumulated heat stress (water quality, salinity, disease, etc.), the changes to the DHW algorithm may have allowed these data to become more prominent in the analysis outcomes. Simply reducing the HotSpot threshold to MMM (or <MMM) rather than MMM + 1 °C resulted in up to 6.8% increases in the hit rate, whilst using an accumulation window of 8 weeks instead of 12 weeks maximised this hit rate. Such improvements were also reflected in model comparison metrics from sensitivity–specificity analyses (increased AUC of 0.02) and Bayesian inference (decreased DIC of 36). As we tested over 200 different DHW configurations, there is unlikely to be any typical bleaching event that can be identified using only the improved DHW configuration. This is because many of the DHW configurations are very similar and are hence referred to as the "suite of best-performing" DHW metrics. These were the DHW metrics with an accumulation window of 4–8 weeks and a HotSpot threshold of MMM or lower. Models using the 4–8 week accumulation window generally performed better, reflecting the typical duration of the vast majority of coral bleaching heat stress events to date [1]. Under climate change, however, average sea temperatures and the duration of marine heatwaves are predicted to continue increasing [1,56], meaning in the future, longer DHW accumulation windows may better capture the levels of heat stress relevant to coral bleaching. Given that baselines are shifting throughout biotic and abiotic marine systems and that rates of adaptation to future environmental conditions are yet unknown, the concepts addressed in this study likely need to be revisited in the future at semi-regular intervals to ensure that the DHW product remains as accurate as possible.

### 4.1. Complexities of Coral Bleaching

Coral bleaching is a stress response whereby photosynthetic algal symbionts are lost from the coral host tissues, resulting in the white coral skeleton becoming progressively more visible [13,57]. Given the complexity of this host–symbiont relationship, survey metrics such as "coral bleaching extent" provide limited information from which to infer biological causes. Coral bleaching is affected by numerous biological factors including symbiont community composition and environmental responses (e.g., more or less heat-tolerant algal taxa) [58], host heterotrophy (e.g., reliance on the symbiont) [59], the capacity for acclimation and adaptation both genetic and epigenetic (intra- and inter-generational) [60,61] and coral taxonomy (e.g., different life history strategies) [24,62]. In addition, other environmental factors can influence bleaching responses in corals, such as high solar insolation, cloudiness, winds, tidal extremes, thermal variability, cold water stress and nutrient enrichment [63–69]. Given this suite of biotic and abiotic factors, a perfectly predicting coral bleaching algorithm would need to combine heat stress metrics with other environmental and biological parameters that, in many cases, are often not available at suffi-

cient spatial or temporal scales. NOAA CRW are investigating the potential improvements to DHW via the inclusion of solar insolation with the development of their Light Stress Damage (LSD) satellite-based product [15].

Here, we refined the ability of a common heat stress metric to predict mass coral bleaching. Ideally, such an optimisation study would be based on coral bleaching data that relate to only heat stress-related mechanisms. By filtering the dataset as described, we did our best to achieve this; however, some bleaching observations in the dataset may inevitably have been caused by other biotic or abiotic factors, contributing to the noise in our results. Bleaching observations from surveys may also be subject to other inaccuracies such as the assumption that sampling only part of a reef is representative of the entire reef. Despite these points, the model comparisons performed in this study remain valid as model biases were applied to all models equally. Given these facts, the AUC and hit rate from sensitivity–specificity analyses are unlikely to reflect the absolute accuracy of DHW metrics but rather allow comparisons of relative accuracy to determine optimal HotSpot thresholds and accumulation windows. The optimisation study presented here was performed on a global coral bleaching dataset. For scientists and practitioners aiming to assess global patterns in coral bleaching, we show that bleaching predictions can be improved by computing DHW metrics using an optimal HotSpot threshold of MMM + 0 °C and accumulation window of 8 weeks. Compared to the operational DHW algorithm, which accumulates only HotSpots greater than MMM + 1 °C, the improved algorithm has a lower accumulation threshold. Accordingly, some corals (e.g., certain species, or at certain locations) may undergo bleaching at lower thermal thresholds than previously predicted. This highlights a need for more research on coral stress responses to low-magnitude heat stress. These recommended DHW algorithm refinements are only applicable to global analyses and predictions of mass coral bleaching caused by heat stress. Moreover, it is important to note that the quasi-opportunistic nature of coral bleaching surveys (i.e., monitoring coral bleaching when DHW values are high, indicating high bleaching risk) can lead to a confirmation bias in studies of coral bleaching and heat stress. Monitoring programmes should address this limitation by aiming to survey bleaching more regularly, even when there is no accumulated temperature stress (i.e., DHW = 0).

### 4.2. Global and Regional Scales

A regionally sensitive DHW algorithm would likely improve predictions of mass coral bleaching. For instance, many scientific studies have used variants of the DHW algorithm to better predict coral bleaching in their study site [23–25]. This will likely continue, since oceanographic and climatic systems, coral assemblages and the distribution of algal symbiont taxa vary geographically and at regional scales [58,70,71]. For instance, the thermal regime of the tropical Eastern Pacific is distinct from many other tropical regions, characterised by high variability due to the El Niño Southern Oscillation, with more intense warm water conditions typical of La Niña years compared to El Niño years [70]. Long-term trends in coral coverage from this region, which have remained very stable over the past three decades, are atypical compared to most tropical reefs which have suffered persistent declines [12,31]. Such distinct trends in the tropical Eastern Pacific could be caused by regional adaptation of corals to these highly variable thermal regimes [31]. This is just one example of a region that could benefit from a specific regional DHW optimisation. Notably, the methods applied in this study would be easily adapted to develop such regional DHW products.

### 4.3. Alternate Heat Stress Algorithm Applications

Optimising heat stress metrics for specific purposes could also be useful for other marine systems. Marine heatwaves have contributed to marked ecological disturbances beyond mass coral bleaching and mortality events [53,72,73], yet specific metrics to predict these other disturbances are not often implemented. The northeast Pacific warming event of 2013–2015, termed "the blob", was the subject of unusually high SST anomalies and

repeated marine heatwaves [74]. The blob was associated with considerable ecological impacts, including the mass stranding of marine mammals such as sea lions and whales [8], die-offs and reproductive failure of seabird populations [6–8] and reduced survival and growth of foraging fish [75]. In all these cases, evidence suggested that declines in higher trophic levels were associated not to direct effects of heat stress, but to the cascading effects of heat-mediated declines at lower trophic levels. Reduced abundance and altered composition of zooplankton communities including krill are highly susceptible to heat stress [3–5], which can result in reduced food availability for higher trophic level animals (e.g., Cassin's auklet and Californian sea lion), leading to their emaciation and mortality [8]. The urgency to understand the full extent of ecological impacts associated with marine heatwaves could, in part, be addressed by creating new heat stress indicators that are optimised for specific disturbances using similar methods to those applied here. While this would not allow for rapid response actions to such events, it would guide marine protected area design (i.e., focus on conserving thermal refugia) and inform future projections of marine systems and related policy recommendations.

### 5. Conclusions

The Anthropocene is characterised by shifting baselines of biological communities, loss of biodiversity and increasingly severe and frequent climatic disturbances. Thus, there is a growing need to understand and be able to predict climatic and anthropogenic disturbances on habitats, particularly those that provide key ecosystem services to socio-ecological systems. Here, we fine-tuned a commonly used heat stress algorithm to a specific purpose (i.e., predicting mass coral bleaching) and showed that simple changes (compared to the operational algorithm) can result in a considerable improvement in prediction success. The philosophy behind this optimisation study was to remove prior expectations, run the models and allow the data to reveal the optimal algorithm parameters (HotSpot threshold and accumulation window) for predicting mass coral bleaching globally. In this case, coral bleaching observations were correctly predicted up to 7.9% more often just by reducing the HotSpot threshold and shortening the accumulation window of the $DHW_{test}$ metric. Broadly, improving bleaching prediction success of the operational DHW metric can support stakeholders and end-users such as coral reef managers, inform the design of MPA networks (e.g., including thermal refugia) and provide more accurate information which can lead to better conservation and restoration decision making (shifting valuable coral nurseries during heatwaves, assisting with decisions on when to relocate aquarium-grown corals to the reef, etc.). Fine-tuning DHWs also has potential for other specific systems, such as predicting planktonic shifts and associated impacts on higher trophic levels. Increasingly under climate change, marine heatwaves are shaping species populations, biological food webs and even ecosystem structure and function [12,53,54]. Thus, optimising our predictions of heat stress and the associated ecological impacts will be key to understanding the future of marine ecosystems.

**Supplementary Materials:** The following are available online at www.mdpi.com/article/10.3390/rs13142677/s1, Figure S1: Coral bleaching data filtration, Figure S2: Model validation, Figure S3: Patchiness simulation test, Figure S4: Best-performing spatiotemporal GLM example—estimated posterior distributions, Figure S5: Best-performing spatiotemporal GLM example—spatiotemporal correlation. Figure S6: Best-performing Spatiotemporal GLM example—Estimated Posterior Distributions. Figure S7: Best-performing Spatiotemporal GLM example—Spatiotemporal Correlation.

**Author Contributions:** Conceptualisation, L.L., J.C.B., A.J.E., J.R.G. and W.J.S.; methodology L.L., P.J.M. and J.R.G.; formal analysis, L.L.; data curation, B.L.S., L.L. and W.J.S.; writing—original draft preparation, L.L. and B.L.S.; writing—review and editing, J.C.B., H.K.E., A.J.E., J.R.G., P.J.M., W.J.S. and B.L.S.; visualisation, L.L.; supervision, J.R.G., J.C.B., H.K.E. and P.J.M. All authors have read and agreed to the published version of the manuscript.

**Funding:** This research was funded by the Natural Environment Research Council's ONE Planet Doctoral Training Partnership (NE/S007512/1) to L.L., the European Research Council Horizon 2020 project CORALASSIST (project number 725848) to J.R.G. and A.J.E. Coral Reef Watch and ReefSense staff (B.L.S. and W.J.S.) were supported by NOAA grant NA19NES4320002 (Cooperative Institute for Satellite Earth System Studies) at the University of Maryland/ESSIC.

**Institutional Review Board Statement:** Not applicable.

**Informed Consent Statement:** Not applicable.

**Data Availability Statement:** The sea surface temperature data that support these findings (Coral-Temp v3.1) are openly available at https://coralreefwatch.noaa.gov/ accessed on 13/05/2020. The coral bleaching dataset that supports these findings will be available at https://www.ncei.noaa.gov/archive/accession/0228498 but is not yet publicly available due to privacy restrictions. All code will be posted to https://github.com/liamlachs upon publication.

**Acknowledgments:** The authors would also like to thank Adriana Humanes and Stephen Rushton for their intellectual contributions to the study and methodology, and the innumerable divers, volunteers and citizen scientists who helped with coral bleaching data collection. The scientific results and conclusions, as well as any views or opinions expressed herein, are those of the author(s) and do not necessarily reflect the views of NOAA or the Department of Commerce.

**Conflicts of Interest:** The authors declare no conflicts of interest.

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
