# Peer review of "Fine-Tuning Heat Stress Algorithms to Optimise Global Predictions of Mass Coral Bleaching"

_remotesensing, doi:10.3390/rs13142677_

Round 1

Reviewer 1 Report

The Manuscript from Lachs et al., concers a methodological approach to estimate global coral belaching events. Authors optimized and applied a model for the estimation of heat stress for prediting global mass bleaching events. The application of the upgraded model will be useful for stakeholders and sea-managers for managing MPAs, monitoring and recovery programmes.

As long as the subject of this manuscript falls within the scope of Remote Sensing, I see no problem.

The manuscript is acceptable in the present form.

Author Response

We would like to thank Reviewer 1 for the supportive and encouraging comments.

Reviewer 2 Report

Degree Heating Weeks (DHW) is a satellite-based heat-stress metrics of coral bleaching that is widely used by researchers and managers. Though its effectiveness has generally been accepted, the configuration to calculate DHW was determined in the early stage of operation and has not been changed based on relationship between heat anomalies and actual records of bleaching. This study reviewed DHW with a global bleaching dataset and fine-scale satellite SSTs, and proposed an improved DHW configuration for better prediction. The statistical methodologies are detailed and sound and their proposed improvement seems to be valid, and is worth to be published in this journal. I hope the authors would take into consideration the comments below.

The results are only shown by the outcome of statistical analysis, and rather hard to get a concrete image for readers who are not familiar with statistics. It would be better to show some examples of typical bleaching events, which can be identified only by the improved DHW configuration.

Their proposed configuration of adding weekly anomalies exceeding MMM over 8 weeks instead of only for the weeks >+1 °C MMM over 12 weeks increased the coral bleaching hit-rate by up to 7.9% (lines 425-427), as the new configuration increased the DHW values to cover more bleaching events. On the other hand, it also increased the false negative rate of 3% (line 429). This negative effect, though canceled out by the positive effect, drives practical problems, and thus should be shown more clearly and evaluated. The DHW threshold values might be changed and should be discussed explicitly.

The original DHW configuration of adding only for the weeks +1 °C MMM is based on an assumption that anomalies <1°C SST are insufficient to cause visible stress to corals. The main result of this study that incorporation of the anomalies <1°C SST better predicts the bleaching may have some implication for physiological response of corals.

Monthly Mean SSTs (MMM) at each site is the fundamental values, based on which the DHW is calculated. MMMs are derived by referring to Skirving et al. (2020) [ref. 19]. The MMMs is fundamental for DHW calculation, so it should be evaluated in detail, or at least procedure to determine MMMs should be explained.

In the section of future outlook, I would like to see discussion how the frequency and severity would be projected by the new DHW under RCP 2.6 or 8.5 on an assumption that corals would not adapt to heat stress, and respond to heat anomalies with the same manner as the present DHW.

Refs 25 and 68 are the same.

Recent studies evaluating DHW by the actual bleaching events are referred to in this study [refs. 23-27]. More early-stage evaluation papers (some are listed below) on DHW are missing but must be shown, as the subject of this paper is DHW (many other papers that are not directly related to DHW are cited).

Liu G, Skirving WJ, Strong AE (2003) Remote sensing of sea surface temperatures during 2002 barrier reef coral bleaching. Eos (Washington DC) 84:137–144

Liu G, Strong AE, Skirving WJ, Arzayus LF (2006) Overview of NOAA coral reef watch program’s near-real-time satellite global coral bleaching monitoring activities. In: Proc 10th int coral reef symp, pp 1783–1793

Liu G, Rauenzahn JL, Heron SF, Eakin CM, Skirving WJ, Christensen TRL, Strong AE (2013) NOAA coral reef watch 50 km satellite sea surface temperature-based decision support system for coral bleaching management. NOAA Technical Report NESDIS 143, NOAA, Washington DC, 33 pp

Strong AE, Liu G, Kimura T, Yamano H, Tsuchiya M, Kakuma S, van Woesik R (2002) Detecting and monitoring 2001 coral reef bleaching events in Ryukyu Islands, Japan using satellite bleaching HotSpot remote sensing technique. Geoscience and Remote Sensing Symposium, 2002. IEEE International 1:237–239

Eakin CM, Morgan JA, Heron SF, Smith TB, Liu G, Alvarez-Filip L, Baca B, Bartels E, Bastidas C, Bouchon C, Brandt M, Bruckner AW, Bunkley-Williams L, Cameron A, Causey BD, Chiappone M, Christensen TRL, Crabbe MJC, Day O, de la Guardia E, Diaz-Pulido G, DiResta D, Gil-Agudelo DL, Gilliam DS, Ginsburg RN, Gore S, Guzman HM, Hendee JC, Hernandez-Delgado EA, Husain E, Jeffrey CFG, Jones RJ, Jordan-Dahlgren E, Kaufman LS, Kline DI, Kramer PA, Lang JC, Lirman D, Mallela J, Manfrino C, Marechal JP, Marks K, Mihaly J, Miller WJ, Mueller EM, Muller EM, Toro CAO, Oxenford HA, Ponce- Taylor D, Quinn N, Ritchie KB, Rodriguez S, Ramirez AR, Romano S, Samhouri JF, Sanchez JA, Schmahl GP, Shank BV, Skirving WJ, Steiner SCC, Villamizar E, Walsh SM, Walter C, Weil E, Williams EH, Roberson KW, Yusuf Y (2010) Caribbean corals in crisis: record thermal stress, bleaching, and mortality in 2005. PLoS ONE 5:e1396

Kayanne H (2017) Validation of degree heating weeks as a coral bleaching index in the northwestern Pacific. Coral Reefs, 36, 63-70.

Author Response

We would like to thank Reviewer 2 for the supportive and constructive comments which have helped us to improve the manuscript. Please see our full point-by-point responses in the attached word document. Author comments are shown in red font.
